# Small molecule screen in embryonic zebrafish using modular variations to target segmentation

Sandra Richter[1,2,3], Ulrike Schulze[2,3,4], Pavel Tomançak [1,3] & Andrew C. Oates [1,2,3,4,5]

Small molecule in vivo phenotypic screening is used to identify drugs or biological activities by directly assessing effects in intact organisms. However, current screening designs may not exploit the full potential of chemical libraries due to false negatives. Here, we demonstrate a modular small molecule screen in embryonic zebrafish that varies concentration, genotype and timing to target segmentation disorders, birth defects that affect the spinal column. By testing each small molecule in multiple interrelated ways, this screen recovers compounds that a standard screening design would have missed, increasing the hit frequency from the chemical library three-fold. We identify molecular pathways and segmentation phenotypes, which we share in an open-access annotated database. These hits provide insight into human vertebral segmentation disorders and myopathies. This modular screening strategy is applicable to other developmental questions and disease models, highlighting the power of relatively small chemical libraries to accelerate gene discovery and disease study.

[1] Max Planck Institute of Molecular Cell Biology and Genetics, Pfotenhauerstrasse 108, Dresden 01307, Germany. [2] The Francis Crick Institute, 1 Midland Road, London NW1 1AT, UK. [3] The Office of ALBS, 190 Euston Road, London NW1 2EF, UK. [4] Department of Cell and Developmental Biology, University College London, Gower Street, London WC1E 6BT, UK. [5]Present address: Institute of Bioengineering, School of Life Sciences, École Polytechnique Fédérale de Lausanne (EPFL), Lausanne CH-1015, Switzerland. Correspondence and requests for materials should be addressed to A.C.O. (email: andrew.oates@epfl.ch)

Phenotypic small molecule screening is used to study basic biological mechanisms, investigate the molecular and cellular basis for many human disorders, and to identify therapeutic drugs. Over the past decade, zebrafish have emerged as a popular vertebrate phenotypic screening model, with over 60 small molecule screens reported[1]. This reflects the advantage that drug activity and toxicity can be assessed in vivo in zebrafish by simply adding compounds to the water.

Despite this technical ease, phenotypic approaches can often be time consuming due to complicated molecular target validation, and assessing their efficiency remains difficult. While potential hit candidates can be retested to control for false positives, revealing false negatives is far more challenging. Moreover, unknown in vivo working concentrations and different target efficacies of small molecules contribute to the likelihood of missing hits. One option to increase the number of hits is to expand the size and structural diversity of the chemical libraries. However, the cost may be restrictive to all but large companies and/or consortia of laboratories. Taking an alternative approach, here we describe a flexible screen designed to maximize the hits from a starting pool of molecules by modulating treatment conditions rather than expanding the small molecule library.

We used zebrafish embryos to model vertebral malformation disorders, which occur at a frequency of 1/1000 in the human population[1,2]. Bones of the axial skeleton and muscle are produced during embryonic segmentation. This complex, multi-step process includes patterning of the progenitor tissue through genetic oscillations of the segmentation clock network, then morphological formation of somites, and finally somite differentiation into axial bones and muscles[3]. Mutations in several genes affecting the segmentation clock in humans, mouse and zebrafish cause segmentation defects of vertebrae known clinically as spondylocostal dysostosis[4]. However, only a small fraction of the genetic causes of vertebral malformations in humans have been identified.

Traditional screening has isolated relatively few zebrafish mutants that affect segmentation. In large-scale ENU mutagenesis screens, somite and muscle phenotypes made up only 1% of the isolated mutants, in comparison to 8% for either heart or eye defects[5,6]. This low frequency may be partly explained by the lack of an accurate and sensitive read-out for segmentation defects in these screens. Further, redundant genetic control may have masked underlying function: for example, mutations in the segmentation clock genes her1 and hes6 exhibit mild phenotypes when knocked out individually, which might have been overlooked, whereas the double mutant combination strongly disrupts segmentation[7,8]. Such redundancy has been widely observed in segmentation[9–12].

To identify players and pathways in vertebrate embryonic segmentation, we selected a library of ~250 biologically active small molecules previously not known to affect segmentation. We included FDA-approved drugs, as well as less-studied small molecules, almost all of which lack in vivo working concentrations in zebrafish. The most common or "standard" zebrafish small molecule screening design uses a single concentration (10 µM) in a wild type genetic background[1], but how this design might be improved by varying and adding to these basic conditions has yet to be systematically explored. Concentration curves are fundamental in toxicology, and a previous high-throughput screen using an enzymatic assay showed that testing a range of concentrations increased the identification of biologically active compounds[13,14]. More recently, a similar approach was successfully used to screen for reporter gene expression in zebrafish larvae[15,16]. We therefore tested each small molecule at three different concentrations.

Screens for small molecules that suppress mutant zebrafish phenotypes have been successfully carried out[17,18], but suppressor or enhancer screens have not yet been used to identify genes or pathways involved in segmentation. We hypothesized that an appropriately sensitized genetic background to increase the identification of active compounds might be provided by the single mutants of her1 and hes6.

Disruption of critical early developmental events, such as the establishment of the basic body plan and tissue layers by mutations or small molecules can obscure essential activities in later processes such as segmentation. To minimize this possibility we added the compounds after axis specification and gastrulation were completed, just before the formation of the first segment. We scored the effects of this perturbation at the end of segmentation using a high-contrast segment boundary marker that captures defects arising as early as the pre-patterning step[19].

Here we show that a flexible screen design, with modules encompassing a range of small molecule concentration, sensitized genetic background and targeted treatment timing, is a practical way to boost the potential of a relatively small chemical library. Our hit number was increased three-fold relative to the standard screening design by these modules, and we identified players and pathways in somitogenesis and myogenesis with implications for human vertebral segmentation disorders and myopathies. This approach enables small laboratories to maximize power of chemical genetic screening, but could also inform the productivity of larger studies.

## Results

**Screening strategy and analysis**. We assembled a library of 243 biologically active small molecules to test their effects during segmentation. These commercially available molecules were selected with the aim of covering a wide range of cellular processes based on the molecular targets assigned by the supplier (Calbiochem). However, in vivo activity and working concentrations of these molecules for zebrafish embryos were almost entirely unknown. We therefore tested each small molecule at three different concentrations (2, 10, and 50 µM) from 10 hpf to 36 hpf, which covers the developmental interval of segmentation without interfering with early embryonic development. We tested the small molecule library in two wild type experimental replicates, and additionally in two segmentation clock mutant strains, her1 and hes6. In total, 12 different treatments were used for each small molecule (three concentrations×four experiments), resulting in around 3000 different treatments. We revealed the segmental pattern by in situ hybridization to detect xirp2a mRNA, a highly specific myotome boundary marker that reflects the characteristic segment pattern and segment shape at 36 hpf, enabling a detailed analysis of segment boundary defects[19,20]. The screening protocol is illustrated in Fig. 1a.

We used a 12-parameter score to describe each treatment-specific phenotype that was shown by at least three out of five embryos per well. We evaluated six segmentation parameters (secondary tail appendage, myotome boundary defects, anterior, trunk and posterior axial position of defects, and segment shape), and also six general morphology parameters (embryonic development, head development, yolk, dorsal-ventral development, axis elongation, tail shape), which together define a "phenotypic vector" (Fig. 1b, c). This phenotypic vector describes how the treated embryos differ from untreated embryos of the respective genotype, with a maximum score of three for each parameter (Fig. 1d). These vectors are illustrated with a color-coded fingerprint for each treatment (Fig. 1c–e). All phenotypes were scored blindly, with treatments identified only by plate positions. Representative images for all treatments and their corresponding phenotypic vectors were collected in a searchable open-source database (www.segsanddrugs.org).

**Technical reproducibility.** To assess the technical reproducibility of our small molecule screen we compared two wild type replicates (wt_A, wt_B). Positive controls inhibiting γ-secretase (DAPT) and FGFR/VEGFR (SU5402), as well as negative controls (0.5% DMSO) on each screening plate showed expected phenotypes[21–23], within and across different experiments (Fig. 1e). We also compared each data point of all scored parameters and the distance of corresponding phenotypic vectors between wild type replicates, finding a positive correlation with a linear correlation coefficient of $r = 0.63$ ($P \ll 0.001$) of corresponding data points in both replicates. A direct comparison of corresponding phenotypic vectors showed an overall Hamming distance of 0.1 (normalized, maximum difference = 1) between wt_A and wt_B. Combined,

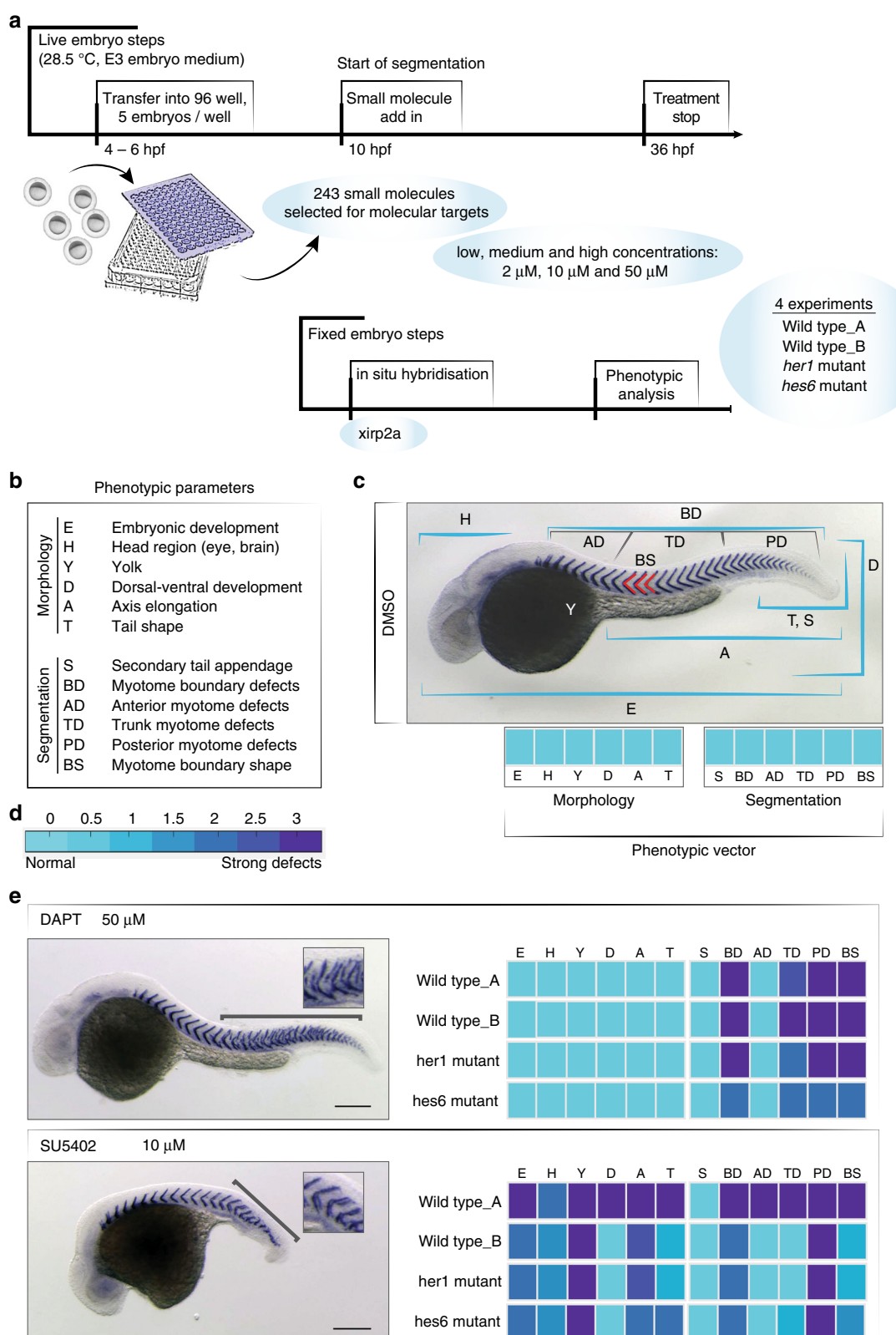

these tests show a high technical reproducibility of the wild type replicates.

**Hit selection based on phenotypic vectors**. Defective segments can be formed as an indirect effect of changes in body shape or elongation. To identify phenotypes of interest that result from a direct effect on segmentation, we sorted the phenotypic vectors to find treatments with high values for the segmentation parameters and low values for the morphology parameters. We term these "direct segmentation" phenotypes, but this does not imply that there are no other changes to the embryo. A good example is DAPT, one of our positive controls (Fig. 1e), which directly perturbs the segmentation clock, but also affects a number of other cell-types such as neuroblasts without detectably changing body shape or elongation[24].

To define hits, we ranked each treatment using the following two methods: (1) by a modified SSMD measurement used for hit selection in RNAi screens[25]; (2) by simply calculating the mean difference of segmentation and morphology parameters. We limited the output to treatments with mean morphology parameter values < 2 and a positive mean difference of segmentation to morphology parameters. These two methods gave highly similar ranking lists (Supplementary Tables 1, 2). On the basis of this output we selected 22 compounds that gave the best direct segmentation phenotypes reproducible in multiple treatments, a hit frequency of 9% of the library (Fig. 2a, Table 1). Because of the overall high reproducibility of wild type replicates, we decided to exclude as false positives seven small molecules with phenotypes observed in only one of the wild type replicates.

**Phenotypic clusters**. The ranking favours segmentation over morphological defects, but does not include information about distinct phenotypic differences across the whole screen. To classify the phenotypes further, we used hierarchical clustering of the phenotypic vectors (Supplementary Fig. 1). Multiple clusters were identified, of which three are described in detail here (highlighted boxes in Fig. 2b; Supplementary Figs. 2–4). The first cluster had the highest ratio of segmentation to morphological defects, which we termed 1st-class phenotypes (orange box, Fig. 2b–d; Table 1). Phenotypes in this cluster show normal axis elongation and segment boundary disruptions with differing spatial restriction along the axis (Fig. 2c, d; Supplementary Fig. 2). The positive control DAPT used at 50 µM[19] appeared in this cluster along with a different γ-secretase inhibitor, Compound E (γ-secretase inhibitor XXI), which had not previously been reported to affect segmentation. Compound E showed a strong posterior trunk and tail segmentation defect similar to DAPT, but had higher efficacy. We found a range of segmentation defects in this cluster, such as defects restricted to the mid-trunk area in wild type treatments with SB225002 (Fig. 2d) or defects all along the axis with XRP44X treatment.

A second cluster was characterized by high scores of "tail shape" and "secondary tail", with the noticeable development of a secondary tail appendage (yellow box, Fig. 2b, e, f; Supplementary Fig. 3). Primary molecular targets listed for small molecules causing this kind of segmentation defect are all non-receptor kinases (JAK3, src, PKR, p38-MAPK and Wee1), which have no previously described function in segmentation.

In a third cluster, we found the positive control SU5402 and other small molecules that affected both segmentation and general development (green box, Fig. 2b, g). Based on the relatively high morphology scores we would have excluded these phenotypes from further consideration, but SU5402 had been reported to cause specific somite defects with shorter time of perturbation[23,26], suggesting that strong perturbation of embryonic morphology may have masked direct segmentation defects in this cluster.

**Shorter time pulse treatment reveals hidden direct segmentation defects**. To investigate whether direct effects on segmentation could be uncovered in the third cluster, we retested a subset of 25 small molecules picked randomly from this cluster, limiting the time of small molecule treatment to a 4 h pulse starting at the onset of segmentation in wild type embryos. Remarkably, direct segmentation phenotypes were observed in seven cases (28%, Supplementary Fig. 5), illustrated by estradiol (Fig. 2g, h). We added these seven small molecules to our hit list (Table 1). These results confirm the general utility of shorter treatment, and suggest that cluster three may harbor considerably more molecules of interest.

**Effect of concentration on direct segmentation phenotype**. Direct segmentation phenotypes were found in all tested concentrations and genotypes. Consistent with toxicity increasing at higher concentrations, embryonic death and overall parameter scores significantly increased with concentration, but remained similar with concentration across all genotypes (Fig. 3a).

To further investigate the relationship between small molecules and phenotypic parameters for the entire screen we used network visualization, as shown for the wt_A experiment in Fig. 3b. Fewer small molecules show defects at 2 µM than at 10 or 50 µM, and stronger defects were found in embryos treated at these higher concentrations as reflected by increased parameter scores. Networks for the other three experiments showed similar results (Supplementary Figs. 6–17). Phenotypes observed with a given small molecule typically varied within an experiment, depending on the concentration. An example is shown with the heat shock protein 90 (Hsp90) inhibitor 17-AAG in Fig. 3c–e, and highlighted in the network visualization (Fig. 3b).

The most commonly used condition reported in zebrafish small molecule screens is 10 µM in wild type embryos[1]. In our screen, choosing either of the two wild type 10 µM replicates gave a hit rate of ~ 3% from the library (Table 1). Across all genotypes, the number of hits increased going from 2 µM (1%) to 10 µM (6%), but were reduced at 50 µM (4%) due to masking by defects in general morphology (Fig. 2a, Table 1). All direct segmentation

**Fig. 1** Screening strategy and phenotypic scoring. **a** Schematic overview of the screening set up. **b** Criteria for phenotypic scoring based on morphological (E, H, Y, D, A, T) and segmentation parameters (S, BD, AD, TD, PD, BS). Each are rated from 0 to 3 (normal to strong difference) compared to untreated controls. All parameters together generate a phenotypic vector for each treatment. **c** Wild type control embryo at 36 hpf, after *xirp2a* in situ hybridization, illustrating the different scoring parameters evaluated on fixed embryos at the stage of 36 h post fertilization (hpf). Segmentation parameters (S, BD, AD, TD, PD, BS) evaluated the quality of formed segments at different axial localization (AD, BD, TD) and overall segment boundary quality (BD), shape (BS) or dorsal-ventral segment defects (S) based on *xirp2a* mRNA expression. The phenotypic vector displayed below the image reflects a normally developed embryo. **d** Numerical parameter score converted into a color scale bar. **e** Representative phenotypes of positive controls: 50 µM DAPT shows posterior myotome boundary defects (bracket and inset) without altering embryonic development otherwise; 10 µM SU5402 affected embryonic development, with a reduction in antero-posterior axis and yolk extension, together with posterior myotome boundary defects (bracket and inset). Associated fingerprints for each positive control illustrate low phenotypic variation in different experiments. Scale bar: 200 µm

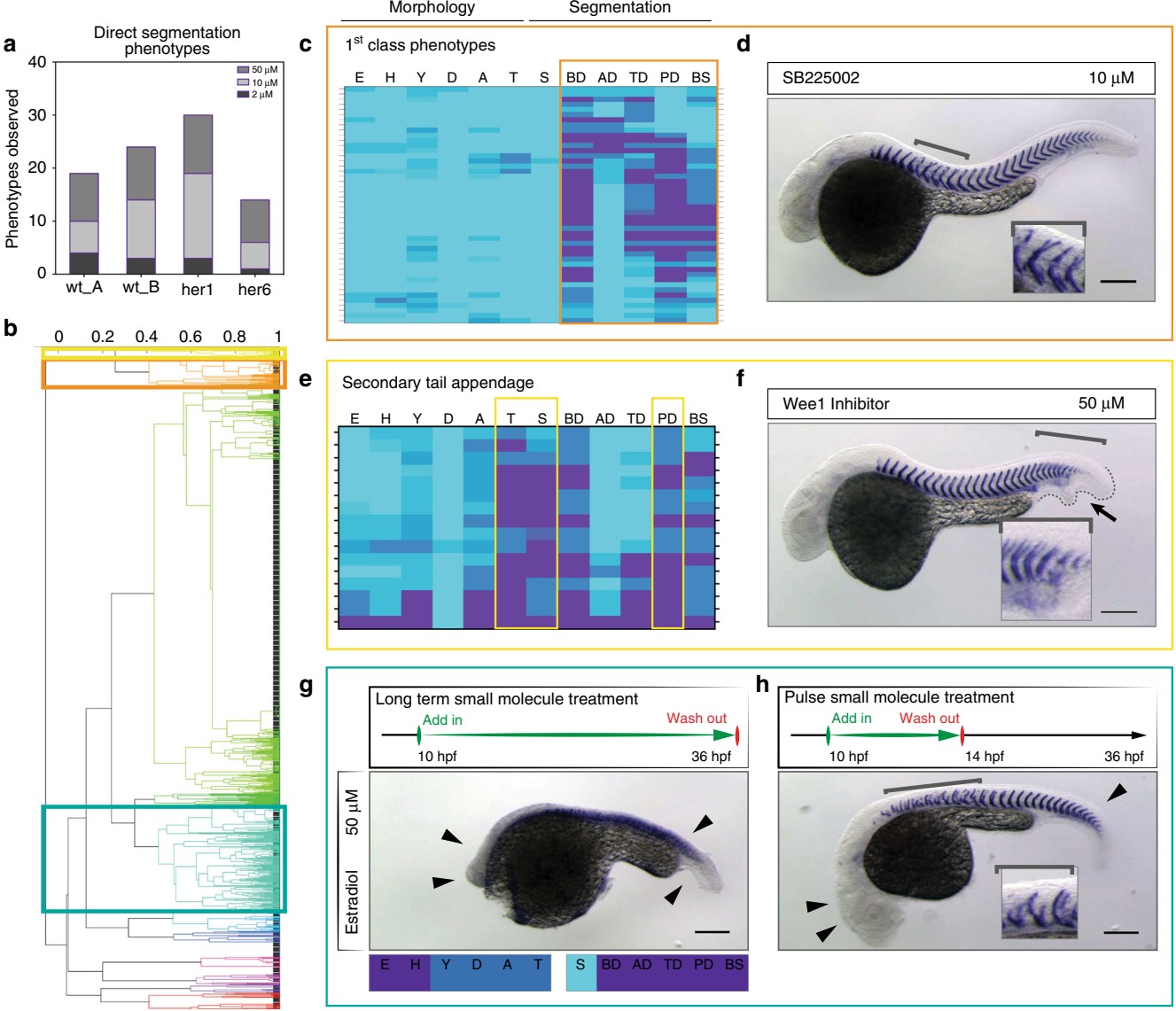

**Fig. 2** Phenotypic hit selection. **a** Number of direct segmentation phenotypes in each experiment identified as hits. The same small molecules may count multiple times, if phenotypes were observed in more than one treatment. **b** Dendrogram from hierarchical cluster analysis of phenotypic vectors. Colored boxes highlight clusters presented in **c**, **e**, **g**; high-resolution dendrogram in Supplementary Fig. 1. Correlation values above plot. **c** Direct segmentation phenotypes identified as "1st class phenotypes", obtained from five different small molecules. **d** 10 μM SB225002 in wild type embryos caused specific mid-trunk defects of myotome boundaries (brackets, inset) with normal embryo morphology. **e** Direct segmentation phenotypes identified as "secondary tail appendage". **f** 50 μM Wee1 Inhibitor in wild type embryos showing relatively normal morphology, but posterior segmentation defects (brackets, inset) with a small secondary tail appendage (arrow, dotted line). **g**, **h** Molecules were selected from cluster with segmentation defects and strong morphological defects. Temporally-controlled pulse in wild type embryos increased segmentation-specificity. **g** Standard long-term small molecule treatment with 50 μM estradiol caused strong developmental defects with rudimentary head and axis in wild type embryos (arrowheads). **h** Pulse experiments of 50 μM estradiol from 10 hpf for four hours showed recovered head and axis structures (arrowheads) and specific myotome boundary disruptions of up to 13 segments (bracket). Inset shows disrupted boundaries. Wild type embryos at 36 hpf, in situ hybridization for *xirp2a*. Scale bar: 200 μm

phenotypes observed at 2 μM and three-quarters of those at 10 μM were masked by strong developmental defects or embryonic lethality at higher concentrations. One-third of hits found at 50 μM were not detected at lower concentrations. Treatment with 2 μM produced the fewest hits in wild type (3), but these were highly reproducible; 10 μM gave the highest hit number in wild type (10), but were the least reproducible between replicates. Importantly, we found that two-thirds of small molecules that produced a direct segmentation phenotype did so at a single concentration.

Altogether, these results show that although the number and variety of morphological phenotypes increased with small

molecule concentration, direct segmentation phenotypes caused by a given small molecule were typically observed at only one of the three tested concentrations.

**Mutant genotype modifies sensitivity for direct segmentation defects.** The *her1* and *hes6* mutant backgrounds gave strikingly different segmentation results compared to wild type and each other (Fig. 4). Untreated, these mutants exhibit relatively normal segmentation. The loss of *her1* causes sparse segment defects up to the 5th anterior boundary. *Hes6* mutants form fewer yet longer segments due to slower somite formation, and occasionally show posterior boundary defects[9,10,27,28].

**Table 1 Small molecules with direct segmentation phenotypes**

| | | 2 µM | | | | 10 µM | | | | 50 µM | | | |
|---|---|---|---|---|---|---|---|---|---|---|---|---|---|
| | | wt_A | wt_B | *her1* | *hes6* | wt_A | wt_B | *her1* | *hes6* | wt_A | wt_B | *her1* | *hes6* |
| *identified independent of genotype* | | | | | | | | | | | | | |
| 1st class phenotypes | | | | | | | | | | | | | |
| XRP44X | Microtubule depolymerization | x | x | x | x | | | | | | | | |
| Compound E | γ-secretase inhibitor | | | | | x | x | x | x | x | x | x | x |
| PD 198306 | MEK1/2 inhibitor | | | | | x | x | x | x | | | | |
| G-1 | GPR30 agonist | | | | | x | x | x | | | | | |
| SB 225002 | Cxcr2 inhibitor | | | | | x | x | x | | | | | |
| Secondary tail appendage phenotypes | | | | | | | | | | | | | |
| Wee1 Inhibitor | Wee1 kinase inhibitor | | | | | | | | | x | x | x | x |
| PP2 | Src kinases inhibitor | | | | | | x | x | x | x | x | x | x |
| SC-68376 | p38 MAPK inhibitor | | | | | | | | | x | x | x | |
| ZM 39923 HCl | JAK3 inhibitor | x | x | x | | | | x | | | | | |
| PKR Inhibitor | PKR inhbitor | x | x | x | | | | | | | | | |
| Segmentation defects with lower specificity | | | | | | | | | | | | | |
| A771726 | DHODH | | | | | | x | x | | x | | | |
| 17-AAG | HSP90 inhibitor | | | | | x | | x | x | | | | |
| Lck Inhibitor | Lck kinase inhibitor | | | | | | x | x | | x | x | x | x |
| Cyclopamine | Smoothened inhibitor | | | | | | x | x | | | x | x | x |
| PD 166866 | FGFR1 inhibitor | | | | | x | | x | | | | | x |
| *identified in the her1 mutant* | | | | | | | | | | | | | |
| IPA-3 | PAK1/2/3 inhibitor | | | | | | | x | | | | x | |
| Baicalein | Lipoxygenase inhibitor | | | | | | | x | | | | | |
| GW9508 | GPR40 agonist | | | | | | | x | | | | | |
| SecinH3 | GEF inhibitor | | | | | | | x | | | | | |
| TBCA | Casein kinase II inhibitor | | | | | | | x | | | | | |
| Heat Shock Protein Inhibitor I | HSF1 inhibitor | | | | | | | | | | | x | |
| STO-609 | CaM-K kinase inhibitor | | | | | | | | | | | x | |

*identified from pulse in wild type*

| | | 10 µM | 50 µM |
|---|---|---|---|
| Estradiol | Steroid hormone | | * |
| LY-83583 | Guanylate cyclase inhibitor | * | |
| Cdk/Crk Inhibitor | Cell cycle kinase inhibitor | * | |
| GW9508 | GPR40 agonist | | * |
| 1-Azakenpaullone | GSK3beta inhibitor | | * |
| GW-5074 | Raf-1 inhibitor | | * |
| Phorbol 12-myristate 13-acetate | PKC activator | | * |

Presence of direct segmentation phenotype marked with x. Small molecules identified after 4h-pulse treatment at indicated concentration (*). Headings and subheadings group small molecules according to our classification. Full details of molecules found in Supplementary Data 1. Gray filling indicates lethal treatments.

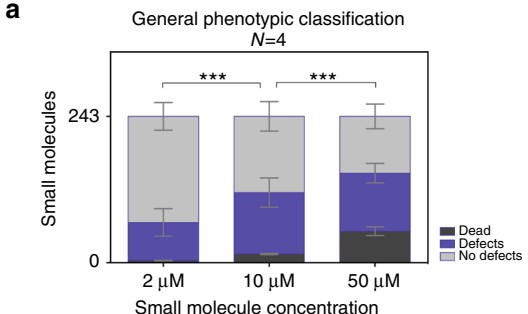

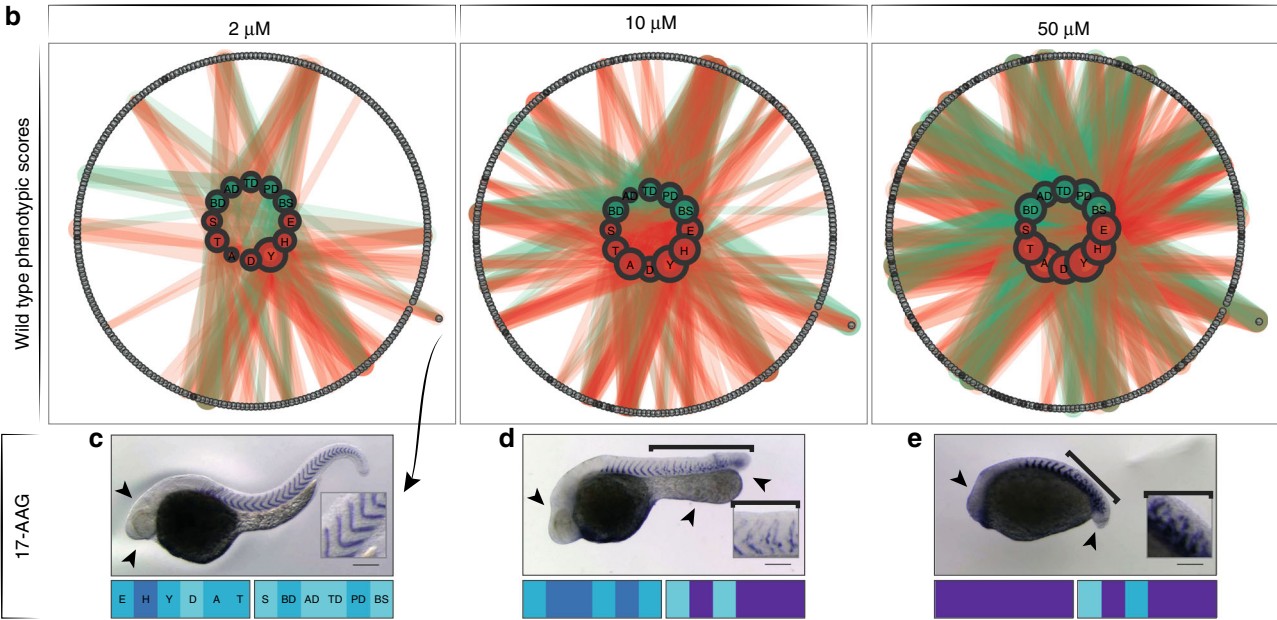

**Fig. 3** Concentration dependence of phenotype. **a** Plot showing the number of lethal treatments, treatments showing at least one defect (any parameter > 0), or no defects depending on the concentration as a mean (± SD) of four independent experiments (2 wild type replicates, 2 mutant genotypes). Lethal conditions significantly increased with increasing concentration (2-way ANOVA, P < 0.0001). **b** Wild type_A phenotypic scores across the concentration series illustrated as interaction networks. Inner circles represent phenotypic parameters: morphology in green, segmentation in red. Size of nodes represents the frequency of scoring > 0 for each individual parameter. Interactions to outer circle represent the link of phenotypic parameter to given small molecule (gray circles). Thickness of interaction lines gives magnitude of score. Dark gray circles, lethal treatments. Small molecule names corresponding to numbers can be found in Supplementary Table 1. Small molecule highlighted in outer circle across concentration series is 17-AAG. Representative images shown in **c–e**. **c** 2 μM 17-AAG, normal segmentation (inset), mild defects of head (arrowheads) and yolk extension. **d** 10 μM, defects of the head morphology (arrowhead) and thickened yolk extension (arrowheads), truncated axis and disrupted myotome boundaries posterior to the 8th (bracket). **e** 50 μM, 17-AAG, strong developmental defects (arrowheads) and disrupted segmentation (bracket, inset). Corresponding phenotypic vector fingerprints below each image show increasing scores for almost all parameters with increasing concentration. Scale bar: 200 μm

We identified seven small molecules that cause direct segmentation phenotypes only in the *her1* mutant (Table 1; Supplementary Fig. 18). For example, the p21-activated kinase inhibitor IPA-3 caused segmentation defects without affecting embryonic morphology at 10 μM (Fig. 4g). Furthermore, *her1* mutants often showed more severe segment defects than wild type replicates of the same treatment (Fig. 4i–k). Thus, the *her1* mutation behaved as an enhancer of small molecule effects. In contrast, we did not identify additional hits in the *hes6* mutant. In fact, 13 direct segmentation phenotypes observed in wild type and/or *her1* mutant were not seen in the *hes6* mutant (Fig. 4d, h, l). General morphological parameters in *hes6* were comparable to the other genotypes, indicating the *hes6* mutant is generally susceptible to small molecule perturbations. The *hes6* mutant retained its characteristic segmentation period phenotype, showing fewer segments than wild type in all conditions where segments could be clearly identified[28]. Indeed, we did not find any treatment with an altered segment number that implied a

change in the rate of segmentation. Thus, the *her1* mutant background was an advantage in identifying small molecules, whereas *hes6* was not.

**Two small molecules affect segments downstream of segmentation clock.** Two inhibitors from the 1st-class cluster, SB225002 and XRP44X, gave striking phenotypes and were examined further (Fig. 5a–c). SB225002 caused boundary disruptions restricted to the mid-trunk segments under wild type screening conditions. A similar spatial restriction has been observed only in *tbx6*/+; *deltaD*/+;*notch1a*/+ triple heterozygote embryos[11]. XRP44X showed mild segment defects evenly scattered along the axis with ectopic *xirp2a* between myotome boundaries. This phenotype is reminiscent of scattered skeletal defects seen in mice with engineered modifications to segmentation clock gene expression[29,30]. Despite segmentation defects, we observed no other changes with SB225002 and only mild developmental defects with XRP44X

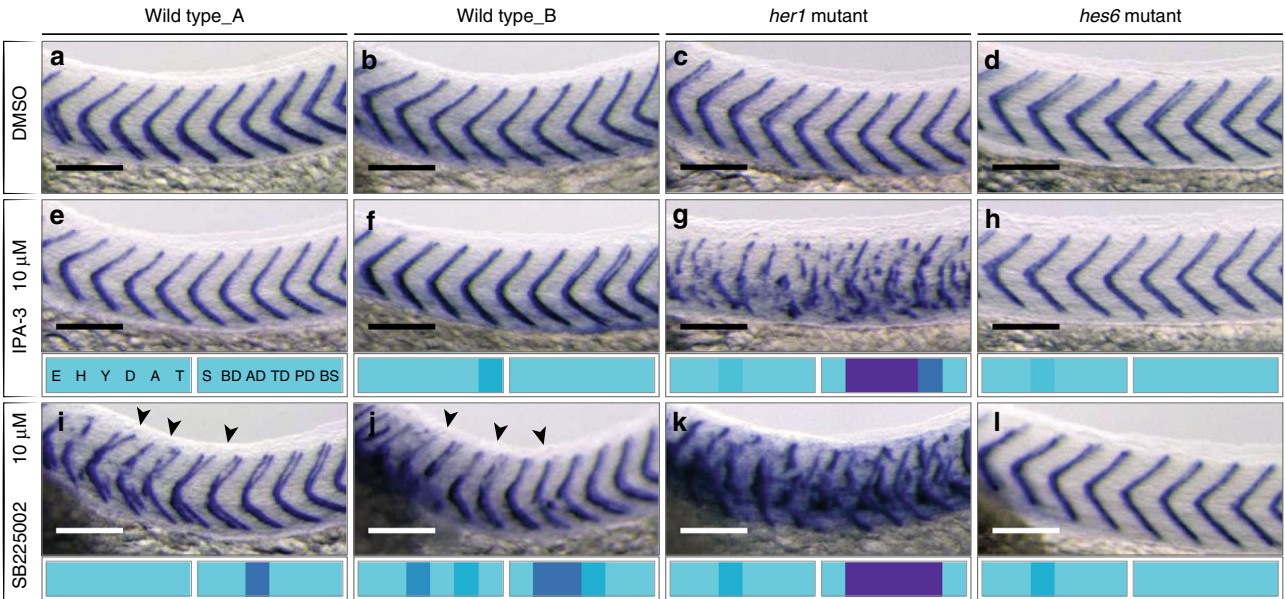

**Fig. 4** Mutant background changed sensitivity to small molecule perturbation. Wild type, *her1* and *hes6* mutant embryos at 36 hpf after small molecule treatment as indicated, *xirp2a* expression in myotome boundaries shown for segments 5–15. **a–d** Myotome boundaries of each genotype normal in 0.5% DMSO control. 10 μM IPA-3 normal segmentation in wild type replicates **e**, **f** and in the *hes6* mutant **h**. **g** 10 μM IPA-3 in *her1* mutants, severe myotome boundary disruptions in trunk. **i**, **j** 10 μM SB225002 in wild type, mild myotome boundary defects of boundaries 4–10 (arrowheads). **k** 10 μM SB225002 in *her1* mutant, severe boundary disruptions, defects extended posteriorly along axis. **l** 10 μM SB225002 in *hes6* mutant, no effect on myotome boundary. Embryos had otherwise normal morphology, as indicated by phenotypic vector fingerprints. Scale bar: 100 μm

including aberrant neurite distribution (Supplementary Figs. 19, 20). These inhibitors have not previously been described to function in vertebrate segmentation.

We investigated the time point at which these molecules disrupt segmentation. Failure of the segmentation clock mechanism can cause segment boundary defects[3,31,32]. We therefore first checked for a loss of the underlying oscillatory pattern by visualizing mRNA expression of the core cyclic gene *her7* in the PSM. *Her7* mRNA was normally expressed (Fig. 5d–f) in embryos treated with either inhibitor, indicating the segmentation clock was unperturbed by SB225002 and XRP44X treatments.

Next, we examined the morphology of the newly formed somites in live embryos during the small molecule treatment. Boundary formation of somites between numbers 6 and 12 was impaired in embryos treated with SB225002, corresponding to the defective myotome boundaries observed under screening conditions (Fig. 5h). In contrast, treatment with XRP44X did not affect initial somite boundary formation (Fig. 5i), indicating that the molecule perturbs a subsequent process.

Finally, we investigated the relationship between the time of treatment and the spatial location of the defects. Instead of adding the small molecules at the onset of segmentation as above, we incubated the embryos from 10 ss until 36 hpf. Treatment with SB225002 resulted in normal segmentation (Fig. 5k), suggesting that SB225002 affects somites at or before their formation, and that the affected segments are sensitive because of their axial location. In contrast, the later add-in of XRP44X caused defects to the myotomes in the anterior trunk, corresponding to somite boundaries that were already formed before treatment started (Fig. 5l). Thus, XRP44X affects maturation of the somites into larval myotomes all along the axis.

**Validation of XRP44X as a microtubule inhibitor during segmentation.** SB22502 has been previously demonstrated to inhibit chemokine receptor Cxcr2 function in zebrafish[33–35], but the mechanism of action of XRP44X is not yet clear. In cell culture,

XRP44X can affect the Ras-Elk signaling pathway, and also inhibits microtubule polymerization with effects similar to combretastatin A4[36]. To investigate possible effects of XRP44X on microtubules in our assay, we first treated zebrafish embryos during mitotic cleavage at around 2 hpf. At these early stages, the embryos have not yet developed a surrounding squamous epithelial barrier layer known as the EVL, and their cytoskeleton is rapidly perturbed by small molecules[37,38]. After 30 min of XRP44X treatment, mitotic spindles were disorganized and reduced in length, and cells were larger and fewer when compared to control embryos of the same developmental timing (Fig. 6a, b). This demonstrates a mitotic arrest, and shows that XRP44X can affect microtubules in zebrafish consistent with its activity as an inhibitor of polymerization in vitro[36].

We next examined the organization of the myotomes of embryos treated with XRP44X in our screening assay by staining F-actin with phalloidin. In the presence of XRP44X, previously formed somite boundaries appear to partially detach and disintegrate with maturation, generating two defective neighboring borders rather than one precise shared boundary (Fig. 6f, g). In untreated embryos, differentiating muscle cells quickly elongated and spanned each newly formed myotome, whereas after XRP44X treatment, these cells were shorter, often failing to span the segment (Fig. 6i, j). In the mature myotome, muscle fibers spanned each segment making contact at each end with the vertical myoseptum, but in treated embryos muscle fibers were disorganized, appearing shorter and curled up (Fig. 6l, m). Treatment with the known microtubule deploymerizing agent combretastatin A4 gave similar results (Fig. 6e, h, k, n). Combined, these findings suggest that XRP44X affects segmentation through its activity on microtubule stability during somite maintenance and muscle differentiation.

In conclusion, these results show that SB225002 and XRP44X affect different stages of segmentation downstream of the segmentation clock: SB225002 interferes with the translation of pattern into somites whereas XRP44X interferes during somite maturation producing a dystrophic myotome (Fig. 7).

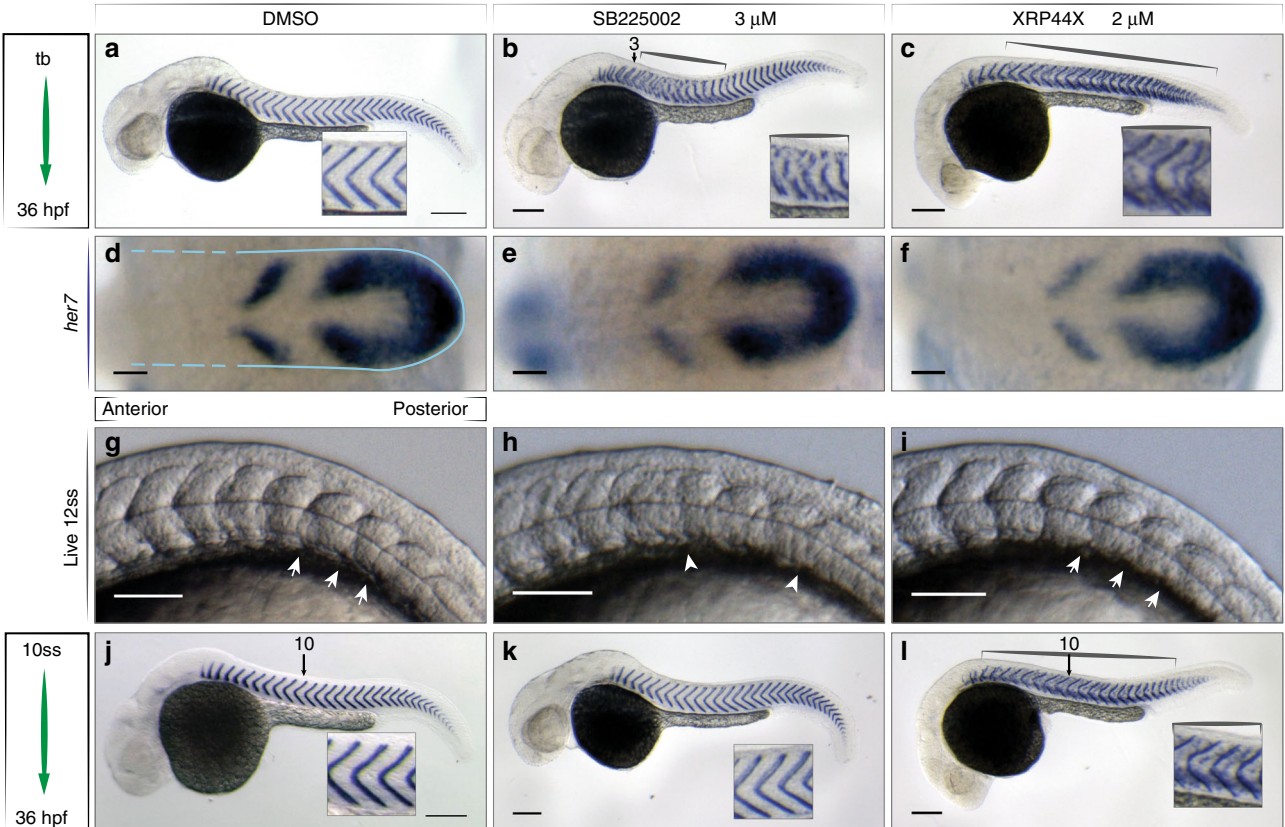

**Fig. 5** Small molecules perturb segmentation at distinct steps. Wild type embryos after in situ hybridization for *xirp2a* mRNA at 36 hpf to mark myotome boundaries **a–c**, **j–l**, *her7* mRNA at 10 ss to show segmentation clock expression **d–f**, and live at 12 ss to show somite boundaries, $N = 3$ **g–i**, treatments as indicated. **a** 0.5% DMSO-treated control embryos. **b** 3 µM SB225002 treatment from tailbud (tb), myotome boundary defects of the mid-trunk, segments 3–15 (brackets, inset). **c** 2 µM XRP44X, segment defects along axis (brackets), ectopic staining of *xirp2a* within myotomes (inset). **d–f** Cyclic *her7* mRNA normal in all treatments. Blue outline indicates PSM, broken lines indicate area of formed somites, ($n = 45$, 3 of 3 experiments). **g** 0.5% DMSO and **i** 2 µM XRP44X showed normal somite formation with intact formed somite boundaries (arrows). **h** 3 µM SB225002, no clear somite boundaries posterior to somite 5 (arrowheads). Last somite boundary not detectable ($n = 30$, 2 of 3 experiments). **j–l** Treatments starting from 10 ss as indicated. **j** Normal segmentation in control. **k** 3 µM SB225002 no effect on myotome boundaries (inset) ($n = 50$, 5 of 7 experiments). **l** 2 µM XRP44X, myotome boundary defects along axis (brackets, insets), affecting segments formed prior to treatment ($n = 40$, 3 of 4 experiments). Scale bar for **a–c**, **j–l** = 200 µm, **d–i** = 50 µm

## Discussion

Using a modular small molecule screening approach, in this study we identified phenotypes and molecular targets in vertebrate segmentation in zebrafish embryos. In total, we found 29 small molecules that appear to affect zebrafish segmentation directly (Table 1). Starting with small molecules previously confirmed to be bioactive, we targeted a specific embryonic developmental time-window, tested multiple concentrations, and made use of segmentation-sensitized genetic backgrounds.

Phenotypic output was strongly affected by the small molecule concentration independent of the genotype. Highly efficacious small molecules identified in our screen at 2 µM were masked by strong developmental defects or early embryonic lethality at higher concentration. It is likely that most molecules reached chemical saturation between 10 µM and 50 µM, causing higher heterogeneity of phenotypes around 10 µM, with loss of specificity at 50 µM due to toxicity interfering with overall embryonic development. Shortening the duration of small molecule perturbation using a pulse delivery reduced these strong morphological developmental effects and revealed direct segmentation phenotypes that would otherwise have been masked.

Pioneering screens to find small molecules that suppress mutant zebrafish phenotypes yielded suppressor hits at frequencies of 1/500-1/2500[17,18]. Out of the two potentially sensitized genetic backgrounds used here, only the *her1* mutant

enhanced segmentation defects. The sensitized *her1* mutant background revealed the highest number of direct segmentation phenotypes of all four experiments (22, a frequency of ~1/11 molecules from the library), including seven that were only active in this background. For example, STO-609 and Casein kinase II Inhibitor III (TBCA) showed comparable phenotypes with posterior segment boundary defects in the *her1* mutant only (Supplementary Fig. 2d). Many small molecules bind to different targets with a range of affinities, and we cannot tell whether the primary target, or targets with a lower affinity were affected in our assay. Nevertheless, according to a comprehensive study of kinase inhibitor selectivity, both these inhibitors are highly effective against Casein Kinase 2 (CK2)[39]. CK2 has been shown to regulate the rhythm of the circadian clock in invertebrates and mammals[40–42], although we did not find evidence for a change in the rhythm of the segmentation clock. The potentiating effect of the *her1* mutant in our small molecule assay may be analogous to the combination of genetic risk and environmental perturbations on vertebral defects seen in heterozygous *Notch* mutant mice exposed to hypoxia[43]. Thus, the *her1* mutant is a promising background to further understand the genotype-environment interaction in segmentation disorders.

In contrast, the period mutant *hes6* suppressed several direct segmentation phenotypes observed in wild type and *her1* mutant embryos. On the basis of the topology of dimerization

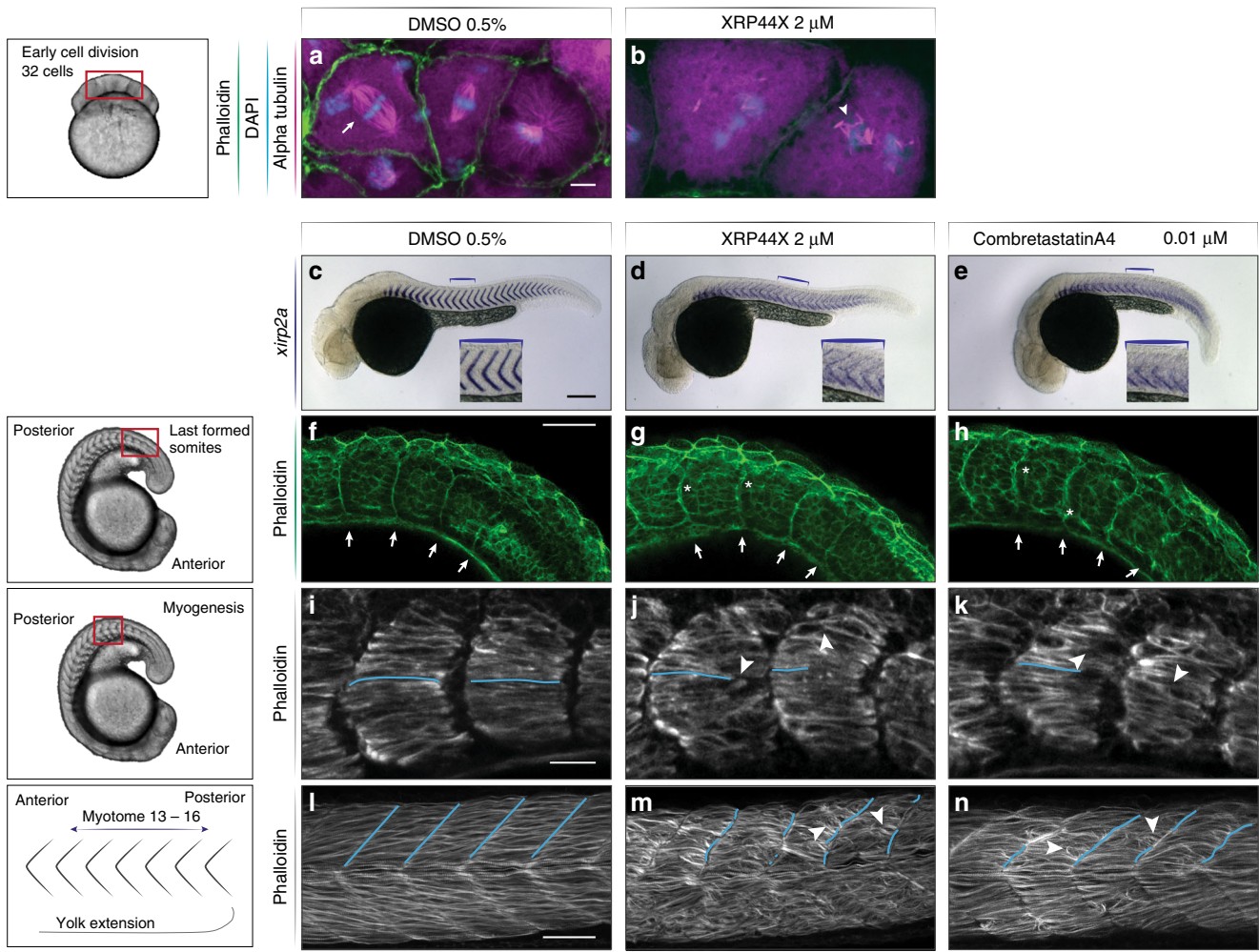

**Fig. 6** Activity of XRP44X on microtubules during zebrafish segmentation. **a**, **b** Zebrafish blastomere cleavage at the 32 cell stage, F-actin in green (phalloidin), alpha tubulin in magenta and DNA with DAPI. **a** DMSO control, normal division with mitotic spindles (arrow). **b** XRP44X treated embryos, short microtubules, no mitotic spindle (arrowhead). Scale bar = 10 µm. **c–e** Embryos at 36 hpf, showing DMSO control, 2 µM XRP44X and 0.01 µM combretastatin A4 treatment. *Xirp2a* mRNA shows myotome boundaries with chevron pattern in DMSO (**c**; inset), ectopic *xirp2a* staining between faint and blurry boundaries with small molecule treatments (**d**, **e**; inset). **f–h** F-actin (phalloidin, green) in 20 ss embryos showing most recently formed boundaries (arrows). **f** In DMSO, thin straight boundary lines form between somites. With XRP44X **g** and combretastatin A4 **h** treatments, boundaries formed less clearly, and mature boundaries had gaps (asterisks). **i–k** During myogenesis, muscle fibers elongate (blue lines) to span the myotome **i**. With XRP44X and combretastatin A4, muscle fibers do not fully span the myotome **j**, **k** remaining partly rounded (arrowheads). **l**, **m** Myotomes of DMSO control and small molecule-treated embryos at 36 hpf. F-actin visualized with phalloidin (gray). Muscle fibers are well organized within each myotome in DMSO **l**, vertical myotome boundaries are highlighted as blue lines. **m**, **n** Small molecule-treated embryos showed loss of muscle fiber structure with round and short muscle fibers (arrowheads) and no clear myotome separation. XRP44X **m** treatment showed a higher level of disorganization than combretastatin A4 **n**. **c–h**, **l–n** scale bar = 50 µm, **i–k** scale bar = 20 µm. N = 30, 3 of 3 experiments showed representative phenotypes

interactions, Hes6 has been previously proposed to act as a "hub" protein in the segmentation clock network[44], but it is not clear why the loss of a hub would cause a less sensitive background[45]. Another hypothesis to explain this surprising result may be that the slower period of somitogenesis in the *hes6* mutant allows enough time between the formation of each boundary to locally repair errors caused by small molecule perturbation. The *hes6* mutant is temperature sensitive, with a more expressive and penetrant phenotype at temperatures colder than used in the screen[28,46]. Repeating the screen at a different temperature (or other environmental variable) may turn up a different spectrum of interactions and thus could be considered as another design module. Whatever the mechanism for the difference in sensitivity between the two mutants, our findings suggest that identification of an appropriately sensitized mutant in pilot experiments is an important part of screen design.

Importantly, the combination of multiple modules into one screening assay increased the harvest of phenotypes from our small molecule library without reducing phenotypic stringency. The effects of these hits were validated by repetition within the assay, and given biological context by cross-reference to enhancement or suppression in the mutant backgrounds. We note that the number of hits in treatments with 2 µM concentration or the *hes6* mutant was very low. Given that the time to screen scaled linearly with the number of treatments, including this concentration and mutant reduced the overall efficiency of the screen in terms of time and effort. The modular design does not necessarily guarantee a more efficient screen than could potentially be achieved simply by increasing the number of compounds. Rather, its strength in reducing false negatives comes from ensuring that each molecule was tested in a range of situations; specifically, out of a total of 29, we uncovered 20 hits

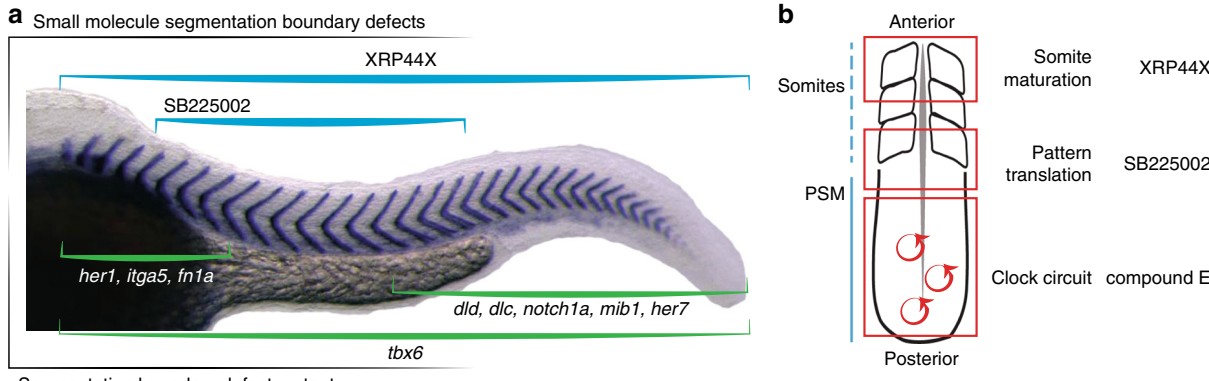

**Fig. 7** Overview of selected direct segmentation phenotypes. **a** Schematic of axial distributions of known mutant somite boundary defects (green brackets) and newly identified small molecule perturbation defects (blue brackets, this paper). Her1, integrin α5 [itga5] and fibronectin 1a [fn1a] anterior defects; T-box 6 [tbx6] mutants, defects along axis; mutants deltaD [dld], deltaC [dlc], notch1a, mindbomb E3 ubiquitin protein ligase 1 [mib1] and her7, posterior defects[11,12,71-74]. **b** Schematic highlighting steps of segmentation (clock circuit, pattern translation, somite maturation) in red boxes and examples of small molecules from the screen that affect the corresponding step

from our starting library that would otherwise have been missed using the "standard" screen design, effectively tripling the number of hits. This strength will be of particular benefit when a library has compounds that have been pre-validated in some way that is expensive and time-consuming, for example compounds that are already FDA-approved for clinical use.

To make these results readily available, we assembled a searchable on-line database that includes representative images and phenotypic vectors of the affected embryos (www. segsanddrugs.org), as well as links to an open-access small molecule database (Bartunek lab www.probesdrugs.org) to combine phenotypic and chemical data. We have mined this data for direct segmentation defects, but a range of other phenotypes are visible and await further investigation using phenotypic vectors as queries.

Our screen assessed early segment patterning reliant on the segmentation clock, as well as subsequent myotome shape and integrity. We identified a variety of segmentation defects, including several at distinct positions along the axis. Somite defects restricted to the mid-trunk region of the axis were observed with SB225002 and IPA-3. From the primary molecular targets of these hits previously reported in zebrafish and other systems, SB225002 inhibits Cxcr2[34,47] and IPA-3 affects PAK1/2 activity[48]. These targets interact in a common pathway in human cells[49] involved in the regulation of EMT[50,51]. Previously, zebrafish Cxcr2 was implicated in chemokine-directed neutrophil migration[34]. In our assay, Cxcr2 inhibition affected boundary formation, but not oscillating gene expression, which suggests Cxcr2 and PAK constitute a pathway of somite epithelialisation in the anterior PSM in zebrafish. Together, these molecules and their targets may help better understand how information from the segmentation clock is translated first into morphological somite formation and then into segmented bones and muscles of the adult.

In several treatments we observed the development of secondary tails. These were similar to previously observed mutant phenotypes in the BMP signaling pathway[52,53], but were caused by small molecules without previous links to this pathway. The JAK3 Inhibitor VI, PKR Inhibitor and SC-68376 were reported to functionally inhibit a number of similar non-receptor serine/threonine kinases, including cyclin-dependent kinases, though SC-68376 showed lower activity[39]. Accordingly, all three inhibitors caused a similar secondary tail phenotype, but SC-68376 did so only at the highest concentration. Common targets of these inhibitors can now be directly assessed for roles in modulating BMP signaling.

One of the hits may provide insight into a human genetic disorder. Targeting of dihydroorotate dehydrogenase (DHODH), a key step in de novo pyrimidine synthesis, in our screen with A77 1726 (Teriflunomide) caused segmentation defects accompanied by relatively mild perturbation to other embryonic structures (Supplementary Fig. 4). Terifluonimide is also known to target the Aryl Hydrocarbon Receptor, but neither the AHR antagonist CH-223191 in our screen, nor the loss of AHR gene function in the embryo[54] show segmentation defects. DHODH is the genetic lesion in Miller syndrome (MIM 263750), a postaxial acrofacial dysostosis (AFD)[55]. Clinical studies have reported vertebral abnormalities in some individuals with postaxial AFD, but it remained unclear whether such defects were associated with Miller syndrome or were diagnostic for related AFD conditions such as Nager syndrome[56]. Our results support the idea that vertebral abnormalities are a primary defect in Miller syndrome. Developmental defects in zebrafish neural crest from DHODH inhibition have been linked to a reduction in transcriptional elongation as a result of limiting pyrimidine concentration[57], suggesting that this may also underlie the segmentation defects we observed.

Existing zebrafish myopathy models, such as sapje, which carries a mutated dystrophin gene have been used to screen FDA-approved compounds for potential therapeutics[58-60]. However, sapje has normal segment boundary shape up to and including 36 hpf[58]. It was therefore surprising that we identified XRP44X as causing a defect in muscle formation and structure during this time window without any observable perturbation to the segmentation clock or initial somite formation. Thus, XRP44X is acting at a later stage than scattered defects caused by engineered alterations in the mouse segmentation clock[29,30]. XRP44X is reported to inhibit Ras-Elk activation in response to FGF signaling in cell culture, and also to disorganize the cytoskeleton and depolymerize microtubules in vitro through the colchicine-binding site[36]. The phenotype of XRP44X was unlike that of the various FGF pathway inhibitors in our assay, indicating that its effects on muscle formation were not due to interference in the Fgf-Ras-ERK signaling pathway.

We showed that in zebrafish, XRP44X can interfere with spindle formation in early mitosis and that it affects muscle cell elongation and structure in the segmentation assay similarly to combretastatin A4, another microtubule depolymerizing agent and anti-cancer theraputic[61]. Also consistent with an action on microtubules in vivo was the aberrant distribution and shorter length of neurites in XRP44X-treated embryos (Supplementary Figs. 19, 20). A block in mitosis was not apparent in the

segmentation assay, which may be due to the protective barrier effect of the embryonic epidermis present at these later stages. Microtubules are thought to be required for proper muscle differentiation, generating an array of paraxial microtubules aligned with the long axis of the elongating myofibre[62,63]. This structure provides a temporary scaffold during myofibrillogenesis for the orderly assembly of the myosin fibers as an intermediate step in assembling the sarcomere[64,65]. An appealing hypothesis is that XRP44X interferes with this process to produce the observed myopathy. The hits from our screen thus provide a pool of molecules to investigate previously undefined targets in vertebrate patterning, somite formation, myogenesis and related disease mechanisms.

In conclusion, designing small molecule screens to allow the flexible exchange of modules, such as developmental timing, genotypic background and readout, can increase the number of relevant hits from a small library without the financial investment in reagents associated with large-scale screening. Our experience here suggests first to identify an appropriately sensitized genetic background and the appropriate developmental treatment window in a pilot screen. Scaling up, different concentrations should be tested, optimally ranging from medium (10 μM) to high (50 μM). Subsequent use of short pulses can dissociate general from specific developmental effects. Manual analysis of the phenotype was suitable for our purpose, but advanced automated image analysis techniques for zebrafish exist and could further increase throughput[66,67]. Importantly, the modular approach described here allows a small laboratory to achieve cost-effective screening with a limited pool of starting molecules.

## Methods

**Fish care and genotyping**. Wild type and mutant zebrafish were maintained according to standard procedures and embryos obtained by natural spawning[68]. Embryos were incubated in E3 at 28.5 °C at all times. Wild type embryos were obtained from AB x TL wild type crosses. Two populations of wild type embryos were harvested on separate occasions from the same pool of adult parents for the wild type replicate experiments. The ENU-induced $her1^{hu2124}$ mutant and the $hes6^{ZM00283007}$ retroviral insertion mutant have been described previously[8,27,28]. Both lines were repeatedly backcrossed into the AB wild type background described above. Mutants were identified by genotyping of adult fish genomic DNA from fin clip samples according to standard protocols[69]. Genotype-specific primer sequences are listed in Supplementary Table 3.

Zebrafish experimentation was carried out in strict accordance with the ethics and regulations of the Saxonian Ministry of the Environment and Agriculture in Germany under license Az. 74-9165.40-9-2001, and the Home Office in the United Kingdom under project license PPL No. 70/7675.

**Small molecule treatment and technical handling**. Embryos of the appropriate genotype were distributed into MultiScreen-Mesh 96-well plates (Merck Millipore) before 10 h post fertilization (hpf) and incubated in E3 until the small molecule add-in. 243 small molecules were selected from libraries provided by Calbiochem (StemSelect® Small Molecule Regulators 384-well library I, InhibitorSelect™ Library I–III). Pipetting was carried out automatically by the ECHO®500 (Labcyte) to distribute volumes of small molecules into master plates. All working concentrations of 2, 10, and 50 μM, were further diluted from one master plate per experiment for higher accuracy using the Tecan Freedom Evo (Tecan) with a 96-head and directly applied to embryos at 10 hpf at a final volume of 100 μl per well to start the treatment. Propylthiouracil was added at a final concentration of 0.003% to each well at 22 hpf to prevent pigmentation of the embryos.

Pulse experiments started from tail bud stage (10 hpf), small molecules were washed out after four hours and embryos were grown under normal conditions until 36 hpf to analyze their phenotypes.

**In-situ hybridization and immunofluorescence**. In situ hybridization was performed at a hybridization temperature of 62 and 65 °C for SSC-washes with riboprobes against $xirp2a$[20] and $her7$[10]. Whole-mount immunofluorescence was used to visualize F-actin with fluorescently tagged phalloidin (Thermo Fisher Scientific, dilution 1:2000) and antibodies against acetylated tubulin (T6793, Sigma Aldrich) and alpha tubulin (GT114, Genetex), each at a dilution of 1:1000.

**Imaging**. In bright field images, embryos were imaged in whole mount on an Olympus SZX16 stereoscope equipped with a CCD camera (QImaging,

Micropublisher). To ensure uniform orientation, embryos were aligned with one half of the yolk cell resting in conical depressions in a pad of agarose (1% in water) cast in a 50 mm petri dish, fitting only the yolk into these conical depressions.

Fluorescent images were taken using a Zeiss LSM 880 with a single photon scanning confocal system with Quasar detector. Fiji open source image software was used to process images.

**Analysis of phenotypic vectors**. Phenotypes were scored manually. Treatments that led to dead embryos or that did not have a score were removed from the analysis. Matlab was used to compute distances between treatments, using correlation along the phenotypic vector as a metric. Clusters were formed using agglomerative average linkage hierarchical clustering. Clusters were formed with a correlation value cutoff of 0.6. The modified strictly standardized mean difference (SSMD)[70] was computed for each treatment using the equation

$$\text{SSMD} = \frac{\overline{X}_{\text{spec}} - \overline{X}_{\text{general}}}{\sqrt{\left(S_{\text{spec}}^2 + S_{\text{general}}^2\right)}}, \tag{1}$$

where $\overline{X}_{\text{spec}}$ and $\overline{X}_{\text{general}}$ are the means of the segmentation and general morphology parameters for each treatment, respectively. $S_{\text{spec}}^2$ and $S_{\text{general}}^2$ denote the variances of the segmentation and general morphology parameters. Treatments where $\overline{X}_{\text{spec}} < 0.2$ and $\overline{X}_{\text{general}} > 2$ were excluded from the SSMD calculations. Treatments were then sorted according to their SSMD and manually evaluated. In addition, the mean difference

$$\text{mean diff} = \overline{X}_{\text{spec}} - \overline{X}_{\text{morph}} \tag{2}$$

was calculated for treatments where $\overline{X}_{\text{morph}} < 2$. The Hamming distance is the proportion of measurements that differ between replicates, described in values between 0 and 1, with 1 indicating the maximum distance between datasets. Networks were generated using Cytoscape. Significance was calculated based on two-tailed $t$-test analysis.

**Data availability**. The authors declare that all data supporting the findings of this study are available within the article and its Supplementary information files or from the corresponding author upon reasonable request. Representative images for all treatments and their corresponding phenotypic vectors are collected in a searchable open-source database (www.Segsanddrugs.org).

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

## Acknowledgements

We thank the Fish Facility and Technology Development Studio of the MPI-CBG, in particular Evelyn Lehmann, Marc Bickle and Martin Stöter, members of the Oates laboratory for discussion and advice, Laurel Rohde for comments on the manuscript, and Petr Bartunek and Ctibor Škuta of the IMG for discussions and connecting us to the probes and drugs database (www.probes-drugs.org).

## Author contributions

S.R. and A.C.O. conceived the study; S.R. carried out the experiments; S.R., U.S. and A.C.O. analyzed the data; P.T. created the database; S.R. and A.C.O. wrote the manuscript and all authors revised it.

## Additional information

**Competing interests:** The authors declare no competing financial interests.

