## [Peer Review File · Nature Communications]

Reviewers' Comments:

Reviewer #1 (Remarks to the Author)

Richter and colleagues examine whether false negatives in small molecule screens can be reduced by performing a matrix of screening conditions using sensitized genetic backgrounds, a range of small molecule concentrations and different temporal treatment intervals. While the central proposition is self-evident, the study provides clear experimental validation. The authors use the zebrafish embryo as a biological specimen as the zebrafish has been successfully employed in prior chemical screens. The authors more specifically screen for defects in embryonic segmentation which lead to defects of the vertebral column. The data are very nice and the text is clearly written.

1. The difference in sensitization in the *her1* and *hes6* mutant phenotypes is striking. Is this due to the peculiar function of *hes6* in gene network? Does the *her7* mutant, or other segmentation mutants, behave more like the *her1* mutant? In other words, if one were designing such a sensitized screen, should one expect most mutants to behave like *her1* or would one expect the sensitivity of any given mutant to be unique?

2. The discussion in the paragraph starting at line 206 makes the interesting point that specific defects for a given compound are only seen at single concentration. Has this observation been made in other chemical screens? Is it common for chemical screens to use a range of concentrations for each compound?

Additional comments regarding scholarship:

This manuscript is a striking example, in a negative sense, of egregious selective self-citation. On line 66, the authors cite genetic redundancy among segmentation clock genes citing an antisense study from 2006 and a paper from their own lab in 2012. This observation was first demonstrated by Henry et al., 2002 using both a mutant and morpholinos and several other subsequent publications.

Line 249-250. The authors describe the SM225002 phenotype as "unlike any previously reported." *Tbx6/+;deltaD/+;notch1a/+* triple heterozygotes were reported to have segmentation defects in the mid-trunk of zebrafish (Julich 2005, *Dev Cell*).

Anterior specific and intermittent segmentation defects have also been reported in the mouse with modifications in *lfn3* transcription and *hes7* splicing (e.g. Shifley et al 2008. *Development* 135: 899-908; Harima et al., 2013. *Cell Reports* 31:1-7).

Line 260. The authors cite their paper from 2007 as showing the failure of the segmentation clock mechanism causes segment boundary defects. There are at least a dozen papers before 2007 reporting that finding.

In Figure 5e, the authors should use the integrin alpha 5, fibronectin 1a, *deltaD*, *deltaC*, *notch1* and *tbx6* instead of the original mutant names as recommended by the zebrafish nomenclature committee. The authors should follow the naming guidelines. The authors cite papers describing the *her7* and *her1* mutants, which include some of their own papers, but they should likewise cite the papers reporting these mutants and their molecular identities (van Eeden et al 1996, *Development*, 123:153-164; Julich et al 2005, *Dev Cell* 8:575-586; Koshida et al. 2005. *Dev Cell*, 8: 587-598; Holley et al., *Genes & Dev* 14:1678-1690; Julich et al, 2005. *Dev Biol* 286: 391-404; Holley et al 2002. *Development*, 129:1175-1183; Nikaido et al 2002. *Nat Genet* 31:195-199).

Reviewer #2 (Remarks to the Author)

The manuscript entitled 'Modular Small Molecule Screening Identifies Novel Vertebrate Segmentation Defects' is focused on developing tiered screening procedures to identify chemicals that visibly modulate zebrafish phenotypes. In this case, the authors focus specifically on screening for molecules that impact segmentation. The manuscript is overall well written and the experiments are generally well-described and in sufficient detail. The statistical data analysis, and visualizations figures are well detailed making the results easy to comprehend.

The authors propose that this is a novel approach that will increase the "hit" rate of small molecules in zebrafish phenotypic screens by combining susceptibility genotypes, chemical concentration, and chemical addition timing. The toxicology field certainly has demonstrated the obvious importance of concentration and timing in producing phenotypic changes, so this aspect is certainly not novel; there are number of large scale zebrafish screening studies in the literature that demonstrate the impact of chemical concentration on hit call rates. Despite this overstatement, this paper could be of interest to others in the chemical genetic field in zebrafish.

A major concern with this manuscript is that the authors repeatedly state that their screening methods identify "specific segmentation hits" The demonstration of segmentation specificity is insufficient to make this claim. Since segmentation was only scored by in situ labeling of the myotomes, there is little evidence that these phenotypes are specific. Certainly, they identified a number of molecules that impact the myotome boundaries.

Furthermore, the authors assume that the chemicals are modulating the zebrafish orthologous targets of these human drugs. This may not be the case, and validation studies were not completed to illustrate that the phenotypes are produced by modulating these pathways. This reviewer understands that the main focus of this manuscript was to suggest an alternative screening strategy, but without the validation studies it is unclear whether or not this approach is superior. As a case in point, the chemical Teriflunomide (the metabolite of Leflunomide) is known to target the AHR in addition to DHODH) in zebrafish. This possibility is not discussed.

Reviewer #3 (Remarks to the Author)

The present manuscript details a chemical screen approach to find new compounds influencing segmentation in zebrafish. The authors in particular want to address the problem of false negatives in screens conducted on one genetic background using only one concentration of the drugs to be tested.

Instead, the authors here test ~250 drugs on 3 different genotypes, in 3 different concentrations and 2 differently timed treatments. Each drug is therefore tested under 12 different conditions. The stated aim of the authors is to allow smaller laboratories to conduct larger screens. Their own findings have been deposited in an online data base, which is comprehensive, intuitive and useful to researchers in the segmentation field.

The strategy to test drugs on a mutant "sensitised" background is very good, albeit not unique. However, the general criticism on the rationale must be that the screening each drug under 12 different conditions is not per se more manageable or cheaper than screening 12 times as many drugs in one concentration on one genetic strain. For the direct comparison of this new method with more conventional approaches, the time spent to analyse each compound needs to be given and has to be contrasted with the hit rate. In the current manuscript, it appears most hits were actually found using the 10 μ M concentration on wild type embryos (despite the statement that 20 hits would not have found with the "conventional screening" approach).

The concept to retest drugs that have pleiotropic effects at lower concentrations is not unique and is being done for such compounds even in the "one concentration" approach.

The detection of segment boundaries by in situ hybridisation is very clear and the pictures very attractive. However, in situ hybridisation itself adds significant time to the protocol and alternative read outs, for example a transgenic fluorescent marker have not been sufficiently considered. This would likely be a prerequisite for automation of the screening process (mentioned in the conclusions).

False negatives can have a number of reasons, including individual drugs in a library having gone bad, or concentrations tested being out of the active range. In larger libraries than tested here, the risk of missing a particular pathway is minimised by several different compounds acting on the same target.

The uniqueness, novelty, and wide potential for the present approach therefore appear widely overstated.

Further points:

It is not clear how the action of SB225002 and XRP44X have been verified in the fish – for example by use of another compound acting on the same target, or by morpholino/CRISPR, so that the authors wouldn't have to "infer" the action (line 341).

It is not clear how the findings support these two pathways as a scoliosis model (lines 350/351).

Line 362ff: the statement about Miller syndrome would need to be verified by additional methods, by morpholino/CRISPR or other genetic method, since an off target effect of the drug cannot be excluded.

Compound E needs to be named, or if there is a patent pending, this must be disclosed.

Minor comments:

In the database, fish should all be oriented facing left, to allow easier comparison between phenotypes.

Authors' Response to Reviewers' comments:

Reviewer #1 (Remarks to the Author):

Richter and colleagues examine whether false negatives in small molecule screens can be reduced by performing a matrix of screening conditions using sensitized genetic backgrounds, a range of small molecule concentrations and different temporal treatment intervals. While the central proposition is self-evident, the study provides clear experimental validation. The authors use the zebrafish embryo as a biological specimen as the zebrafish has been successfully employed in prior chemical screens. The authors more specifically screen for defects in embryonic segmentation which lead to defects of the vertebral column. The data are very nice and the text is clearly written.

1. The difference in sensitization in the *her1* and *hes6* mutant phenotypes is striking. Is this due to the peculiar function of *hes6* in gene network? Does the *her7* mutant, or other segmentation mutants, behave more like the *her1* mutant? In other words, if one were designing such a sensitized screen, should one expect most mutants to behave like *her1* or would one expect the sensitivity of any given mutant to be unique?

We do not currently understand the difference in sensitization between *her1* and *hes6*. The two lines had been back-crossed onto the same wildtype background for many generations, so I don't think it's a property of a second site mutation. The *hes6* gene has been previously proposed as a "hub" in the segmentation clock gene network by Trofka et al., 2012, however, it is not clear to me why a missing hub protein would give a less sensitized background – my expectation based on network theory is that it would be more sensitive (Albert et al., Nature 2000). There's clearly something interesting going on here, but there's no obvious hint from the small molecule hits and so I argue that this is beyond the scope of the current work. The *hes6* mutant has a more expressive and penetrant phenotype at 22 degrees than at the temperature our screen was performed (28), so it is possible that a repeat of the screen on *hes6* at this temperature would yield a similar number of enhancer hits. We did not examine the *her7* mutant background in screening conditions. I think from our available evidence we have to conclude that in designing such a sensitized screen, one ought to test a range of mutants, if possible, and pick the most sensitive one before scaling up. We now include these ideas in the discussion on lines 379-394.

2. The discussion in the paragraph starting at line 206 makes the interesting point that specific defects for a given compound are only seen at single concentration. Has this observation been made in other chemical screens? Is it common for chemical screens to use a range of concentrations for each compound?

It is indeed common for to use a range of concentrations to determine an IC50, for example, and this is the basis for much of toxicology, as pointed out by reviewer 2. We included this connection to basic toxicology in the

introduction on line 80. Using a range of concentrations in a high-throughput screening approach as was first done by Inglese et al., 2006 on an enzymatic assay, cited in the introduction. Also cited in the introduction, Wang et al., 2015 explicitly use a range of concentrations to screen transgene expression in zebrafish larvae and in this case the output is a single expression variable. However, the phenotypic screens we found in the literature don't make the point clear, because they don't show or discuss phenotypic variation at different concentrations. Thus, we feel that part of our contribution is to point out something that perhaps should have been obvious, but had not been taken advantage of.

Additional comments regarding scholarship:

This manuscript is a striking example, in a negative sense, of egregious selective self-citation.

We apologize for this poor scholarship. It was not our intention to exclude the seminal contributions of others. In some cases these omissions were due to a reference formatting error in the final version, but in others we had missed the information in the original report. We are happy to correct all these instances.

On line 66, the authors cite genetic redundancy among segmentation clock genes citing an antisense study from 2006 and a paper from their own lab in 2012. This observation was first demonstrated by Henry et al., 2002 using both a mutant and morpholinos and several other subsequent publications.

We cite the observed redundancy between *her1* and *hes6* as an example because we will use these genes in the screen - the references are correct. To avoid the impression that this is the only example of genetic redundancy in segmentation, we include the following sentence (lines 69-71): "Such redundancy has been widely observed in segmentation (Henry 2002; Oates and Ho 2002; Julich 2005; Koshida 2005)."

Line 249-250. The authors describe the SM225002 phenotype as "unlike any previously reported." *Tbx6/+;deltaD/+;notch1a/+* triple heterozygotes were reported to have segmentation defects in the mid-trunk of zebrafish (Julich 2005, Dev Cell).

We had overlooked this interesting finding and have now cited it in lines 273-274. We have therefore also removed the claim that the SB225002 is "unlike any previously reported."

Anterior specific and intermittent segmentation defects have also been reported in the mouse with modifications in *lfng* transcription and *hes7* splicing (e.g. Shifley et al 2008. Development 135: 899-908; Harima et al., 2013. Cell Reports 31:1-7).

The reviewer is correct. We had previously omitted these mutants because most of this data comes from skeletal staining in mouse and we were not convinced of the relevance, but we now include them on lines 276-278 in the

results. We return to the difference in mechanism (clock versus microtubules) later in the discussion on lines 469-471.

Line 260. The authors cite their paper from 2007 as showing the failure of the segmentation clock mechanism causes segment boundary defects. There are at least a dozen papers before 2007 reporting that finding.

This is of course correct: we have replaced this erroneous reference with 3 reviews that give a balanced overview of the relationship between segmentation clock failure and boundary defects: Holley 2007; Oates et al., 2012; and Hubaud and Pourquie 2014.

In Figure 5e, the authors should use the integrin alpha 5, fibronectin 1a, deltaD, deltaC, notch1 and tbx6 instead of the original mutant names as recommended by the zebrafish nomenclature committee. The authors should follow the naming guidelines. The authors cite papers describing the her7 and her1 mutants, which include some of their own papers, but they should likewise cite the papers reporting these mutants and their molecular identities (van Eeden et al 1996, Development, 123:153-164; Julich et al 2005, Dev Cell 8:575-586; Koshida et al. 2005. Dev Cell, 8: 587-598; Holley et al., Genes & Dev 14:1678-1690; Julich et al, 2005. Dev Biol 286: 391-404; Holley et al 2002. Development, 129:1175-1183; Nikaido et al 2002. Nat Genet 31:195-199).

We have made the necessary changes to the figure legend of figure 6, where the diagram now resides.

Reviewer #2 (Remarks to the Author):

The manuscript entitled 'Modular Small Molecule Screening Identifies Novel Vertebrate Segmentation Defects' is focused on developing tiered screening procedures to identify chemicals that visibly modulate zebrafish phenotypes. In this case, the authors focus specifically on screening for molecules that impact segmentation. The manuscript is overall well written and the experiments are generally well-described and in sufficient detail. The statistical data analysis, and visualization figures are well detailed making the results easy to comprehend.

The authors propose that this is a novel approach that will increase the "hit" rate of small molecules in zebrafish phenotypic screens by combining susceptibility genotypes, chemical concentration, and chemical addition timing. The toxicology field certainly has demonstrated the obvious importance of concentration and timing in producing phenotypic changes, so this aspect is certainly not novel; there are a number of large scale zebrafish screening studies in the literature that demonstrate the impact of chemical concentration on hit call rates. Despite this overstatement, this paper could be of interest to others in the chemical genetic field in zebrafish.

We are not aware of published large-scale zebrafish screens that have analyzed the impact of chemical concentration on hit rate calls, except for the paper that we cite on line 84 in the introduction (Wang et al., 2015). Nor have we found any that analyze the effect of concentration on the development of the embryo. If the reviewer can identify suitable references, we would be happy to cite them.

A major concern with this manuscript is that the authors repeatedly state that their screening methods identify “specific segmentation hits” The demonstration of segmentation specificity is insufficient to make this claim. Since segmentation was only scored by in situ labeling of the myotomes, there is little evidence that these phenotypes are specific. Certainly, they identified a number of molecules that impact the myotome boundaries.

We thank the reviewer for pointing out this problem, and indeed we agree with their point. It was not our intention to make this strong claim, but rather to distinguish direct from indirect effects on the segments. This is because segments can have defective or altered shapes as a secondary or indirect consequence of changes in the shape or elongation of the body axis, whereas we were only interested in those compounds that gave a direct or primary effect on segmentation. We were not concerned, to first approximation, if the treatment also affected other processes in the embryo, as long as the effect on segmentation was primary.

In this sense, we had intended to describe the specificity of the defects in terms of the values of the phenotypic vectors from the “segmentation” category versus the general “morphology” category. Our approach was to select those treatments that gave high segmentation values and low morphology values and we had termed these “segmentation-specific”.

A good example is the positive control compound DAPT (Figure 1e), which inhibits the gamma-secretase activity required for Notch signaling. Notch signaling is involved in many processes in the embryo (neurogenesis, pronephric differentiation, blood vessel sprouting etc.), yet we know from a large body of literature that Notch signaling acts directly in the segmenting tissue, and indeed the body shape and axial extension are normal. Hits with similar general properties reflected in the ratio of phenotypic vector values would be expected and welcome in such a screen.

We included the basis for selecting these so-called “segmentation-specific” hits on lines 113-119 and 142-147 in the original paper, but it is clear now that this was not well described and open to ambiguities. Indeed, given that the term segmentation-specific also reasonably means that the embryo had no other defects but segmentation defects, it is clearly misleading.

We have done two things to rectify this problem: firstly we have added a paragraph at the beginning of the Phenotypic Hit Selection section in the results (lines 154-162) to clearly outline the goals and terms we will use – we now use the term “direct segmentation” throughout the paper instead of “segmentation-specific”; hopefully this should remove any ambiguity.

Secondly, we have investigated the specificity of the phenotype *sensu strictu* for the two molecules we focused on in the latter section of the results: SB225002 and XRP44X. In two supplementary figures (5 and 6) we now include confocal images of embryos triple-stained with phalloidin to detect actin and reveal cell shape, DAPI to mark the location of nuclei, and acetylated tubulin to outline the distribution of neurites in the embryo. From these images, SB225002 appears to have no other embryological defects, although we know from published literature that neutrophils will be defective because *Cxcr2* is blocked (Deng et al., 2013). In contrast, XRP44X shows several mild morphological defects in its interior structures – including aberrant neurites – which might be expected from a compound that has the ability to affect microtubule stability. We report this in the results on lines 278-281, and we discuss the potential link to microtubule function in the discussion on lines 477-493.

We hope that these changes have corrected the main issue of poor and misleading terminology as well as providing a more direct measure of the specificity of the effects of the two molecules we selected for further validation.

Furthermore, the authors assume that the chemicals are modulating the zebrafish orthologous targets of these human drugs. This may not be the case, and validation studies were not completed to illustrate that the phenotypes are produced by modulating these pathways. This reviewer understands that the main focus of this manuscript was to suggest an alternative screening strategy, but without the validation studies it is unclear whether or not this approach is superior. As a case in point, the chemical Teriflunomide (the metabolite of Leflunomide) is known to target the AHR (in addition to DHODH) in zebrafish. This possibility is not discussed.

We do not assume that the chemicals are modulating the zebrafish orthologs. In the discussion on lines 311-313 (of the previous version) we explicitly stated that we do not know whether the reported primary targets are bound by the small molecules. Nevertheless, the point that validation is a key part of any screening approach is well made and we have now revised the manuscript to improve confidence of the targets of three compounds – SB225002, Teriflunomide, and XRP44X. We argue that validating the full set of targets is beyond the scope of this work.

In our previous version, we stated that *Cxcr2* was involved in leukocyte migration in zebrafish, and also that *Cxcr2* was inhibited by SB225002 in cell culture, but we did not make the connection for the reader that SB225002 has been previously used and validated in zebrafish embryos *in vivo* to inhibit the function of the *Cxcr2* receptor during neutrophil migration (Oliveira et al., *J. Immunol* 2013; Deng et al., *J Leuk Biol* 2013; Toraca et al., *Dev Comp Immunol* 2017). We now draw attention to the precedent of this previous work on lines 307-308.

As part of our screen, we had tested the AHR antagonist CH-223191 (CAS no 301326-22-7), which showed a normal phenotype at all tested concentrations.

Further, work analyzing the function of the 3 zebrafish AHR receptors during embryogenesis using a combination of mutants and morpholino knockdowns found no segmentation defects (Goodale et al., 2012). Combined, this argues strongly that the effects on segmentation we observe are not due to targeting the AHR receptors. We have now included this extra information before the description of Miller syndrome in the discussion on lines 449-451.

XRP44X has been described with two potential mechanisms – it can affect activity in the Ras-Elk3 pathway, although no direct molecular target has been proposed, and it has also been shown to depolymerize microtubules in vitro and in cell culture by direct binding to microtubules (Wasylyk et al., Semenchenco et al., 2016). Validation of XRP44X's action on microtubules was performed by first showing that exposure to XRP44X depolymerized the microtubule arrays during mitosis in the very early cleavage-stage zebrafish embryo. Secondly, we tested the known microtubule depolymerization compound combretastatin A4 in the segmentation screening assay and observed a close match to the phenotype of XRP44X. Combined, these new experiments provide a validation of the activity of XRP44X on microtubules during segmentation.

Given the late-occurring phenotype during muscle differentiation of XRP44X in our assay, these new observations make a striking link to previous work in myogenesis in which a role for parallel bundles of microtubules aligned with the elongating cell act as a template for the assembly of orderly myofibrils and subsequently the formation of the sarcomere (Warren 1974; Antin et al., 1981; Pizon et al., 2005). This XRP44X defect now looks like a potentially interesting model for myofibrillar myopathy. We have made a new figure illustrating these findings and include this in the main text as a new figure 6, with corresponding text in the results on lines 306-335 and discussion on lines 462-493.

We argue that these revisions have provided improved validation of three of the interesting hits, and thus increase confidence that application of this screening strategy can yield a superior outcome to the standard approach.

Reviewer #3 (Remarks to the Author):

The present manuscript details a chemical screen approach to find new compounds influencing segmentation in zebrafish. The authors in particular want to address the problem of false negatives in screens conducted on one genetic background using only one concentration of the drugs to be tested.

Instead, the authors here test ~250 drugs on 3 different genotypes, in 3 different concentrations and 2 differently timed treatments. Each drug is therefore tested under 12 different conditions. The stated aim of the authors is to allow smaller laboratories to conduct larger screens. Their own findings have been deposited in an online data base, which is comprehensive, intuitive and useful to researchers in the segmentation field.

The strategy to test drugs on a mutant “sensitized” background is very good, albeit not unique. However, the general criticism on the rationale must be that the screening each drug under 12 different conditions is not per se more manageable or cheaper than screening 12 times as many drugs in one concentration on one genetic strain. For the direct comparison of this new method with more conventional approaches, the time spent to analyse each compound needs to be given and has to be contrasted with the hit rate. In the current manuscript, it appears most hits were actually found using the 10 μ M concentration on wild type embryos (despite the statement that 20 hits would not have found with the “conventional screening” approach).

The reviewer is correct that sensitized backgrounds have been used before. Indeed, one of the pioneering studies of chemical screening in zebrafish looked for drugs that could suppress the *gridlock* phenotype (Peterson et al., Nat Biotechnol 2004) and another shortly after suppressed the *crash and burn* mutant (Stern et al., Nat Chem Biol 2005). However, we are not aware of a published paper that uses a genetically sensitized background in zebrafish to find new molecules, in genetics terms akin to an “enhancer screen”. If the reviewer can identify enhancer screens that we have overlooked, we would be happy to cite them. We now cite the suppressor screens in the introduction on lines 86-89, and return to them in the discussion on lines 357-359.

The question of being more manageable or cheaper is a good one that we did not address properly; I think there are two points here.

The time to screen goes up essentially linearly with the number of treatments using our protocol, so screening each drug under 12 different conditions is approximately the same effort as screening 12 times as many drugs.

From Table 1, we found 6-8 hits (~3%) using the standard design (depending on which wildtype replicate you choose). In contrast, we found 15 hits (6%) with 10 μ M on the *her1* background, which was the most successful single combination, and in fact all hits (excluding those from the pulse treatment) were picked up at some concentration in *her1*. However, overall this advantage was off-set by two modules with low hit rates: 2 μ M concentration and the *hes6* background. Excluding the standard design, we found an additional 12 hits, and another 7 using the pulse delivery, which is 19/250 ~8%, so about 3x the number. Indeed, we would have missed all these if we had carried out the standard screen. However, screening outside the standard design required 8x the treatments, and so the hit rate on average was in fact 3/8 that of the standard design. So we agree that the time/effort efficiency of our screen in finding new hits was not demonstrably better than standard screening, largely due to the inclusion of the low concentration and the *hes6* mutant. We discuss these issues in a new paragraph on lines 395-411. We have also altered our terminology throughout the paper to talk of hit frequencies (out of the total molecules in the library) versus hit rates (which can also apply to number of treatments screened).

Nevertheless, I think one can still make a strong argument about the cost in

practical terms for small laboratories. A sample of commercially available chemical libraries with known activities range from 2 to 25K, and most chemical libraries contain more than enough reagent to screen multiple times, and so most of the chemical is not used. For large, well-funded labs with multiple technicians and postdocs, the lion's share of the costs is labor, and so buying more compounds is a potentially cost-effective means of finding more hits. For small laboratories, particularly those in a university or liberal arts setting, a significant part of the labor comes through undergrad or masters students, and the cost of the library can be a significant proportion of the lab's budget. For this situation, using a standard screen alone might be a huge waste of the discovery potential of the reagent. This is especially so when the library has been pre-validated in some way – for example, if one considers a library of drugs that are already FDA-approved for use in the clinic, then the cost of adding a single new compound via clinical trials to this library is much higher than doing the screen.

Of course, new compounds need to be screened – no question. The question is who should do it? We argue that the modular screen makes hits more accessible for a small lab, and is an efficient use of limited resources. We mentioned this only briefly in the previous version, but now we have focused this to explain where the benefits versus the standard screen are in the new paragraph (above) and in the final paragraph lines 505-506.

The concept to retest drugs that have pleiotropic effects at lower concentrations is not unique and is being done for such compounds even in the "one concentration" approach.

The detection of segment boundaries by in situ hybridisation is very clear and the pictures very attractive. However, in situ hybridisation itself adds significant time to the protocol and alternative read outs, for example a transgenic fluorescent marker have not been sufficiently considered. This would likely be a prerequisite for automation of the screening process (mentioned in the conclusions).

In a set of pilot experiments we tested a number of markers for the segmental boundaries, including several antibodies with much shorter handling time, and we looked into several transgenic lines. On balance we decided to use the in situ method despite its duration because of its very high contrast, its direct connection to the literature, and because the number of embryos in our analysis although large was still manageable by hand, which did not require any further protocol development. We argue that bringing a transgenic reporter of the segment boundary is beyond the scope of the current work.

False negatives can have a number of reasons, including individual drugs in a library having gone bad, or concentrations tested being out of the active range. In larger libraries than tested here, the risk of missing a particular pathway is minimised by several different compounds acting on the same target.

This is certainly true. We would also argue that an advantage of the modular

screen design is a set of cross-relatable data points with the same molecules – seeing phenotypes repeated as well as being enhanced and suppressed can provide a richer data-set already from the primary screen. We mention this point on line 397-399.

The uniqueness, novelty, and wide potential for the present approach therefore appear widely overstated.

We hope that our responses above and the corresponding changes to the manuscript have made the potential of the present approach clearer.

Further points:

It is not clear how the action of SB225002 and XRP44X have been verified in the fish – for example by use of another compound acting on the same target, or by morpholino/CRISPR, so that the authors wouldn't have to “infer” the action (line 341).

Verification of the actions of the small molecules is an important point. We refer to our response to reviewer 2, above.

It is not clear how the findings support these two pathways as a scoliosis model (lines 350/351).

We have revised this sentence, now on lines 431-434 to read: “Together, these molecules and their targets may help better understand how information from the segmentation clock is translated first into morphological somite formation and then into segmented bones and muscles of the adult.”

Line 362ff: the statement about Miller syndrome would need to be verified by additional methods, by morpholino/CRISPR or other genetic method, since an off target effect of the drug cannot be excluded.

We addressed some aspects of the specificity of the hit for DHODH in Miller syndrome in our response to reviewer 2, above.

Compound E needs to be named, or if there is a patent pending, this must be disclosed.

Compound E is the widely used alternative name for γ -Secretase Inhibitor XXI and is commercially available under both names. We have included “ γ -Secretase Inhibitor XXI” now on lines 183-186.

Minor comments:

In the database, fish should all be oriented facing left, to allow easier comparison between phenotypes.

We re-oriented all the fish.

Reviewers' Comments:

Reviewer #1:

Remarks to the Author:

The authors have satisfactorily addressed my concerns in this revised submission.

Reviewer #2:

Remarks to the Author:

Thank you for responding do my comments thoroughly.

There are several large scale screens conducted across concentration in zebrafish demonstrating the concentration dependency on hit call. A couple example I found Truong et al. 2014 Toxicol Sci. 2014;137(1):212-33 Yozzo KL et al. Environ Sci Technol. 2013;47(19):11302-10.

Reviewer #3:

Remarks to the Author:

In this revised manuscript, the authors have addressed most concerns presented by the three reviewers. In particular, they have moderated and specified their statements on the application of the screen, expanded the literature cited and have presented new data to validate the phenotypes generated by their hit.

In general, this paper presents a refined method for using zebrafish as a model for segmentation defect.

Response to Referees **Richter et al., NCOMMS-16-19399A**

Reviewer #1 (Remarks to the Author):

The authors have satisfactorily addressed my concerns in this revised submission.

No response necessary.

--

Reviewer #2 (Remarks to the Author):

Thank you for responding do my comments thoroughly.

There are several large scale screens conducted across concentration in zebrafish demonstrating the concentration dependency on hit call. A couple example I found

Truong et al. 2014 Toxicol Sci. 2014;137(1):212-33

Yozzo KL et al. Environ Sci Technol. 2013;47(19):11302-10.

Thanks for pointing these out. We have now included these references in the introduction in the second paragraph on page 3.

--

Reviewer #3 (Remarks to the Author):

In this revised manuscript, the authors have addressed most concerns presented by the three reviewers. In particular, they have moderated and specified their statements on the application of the screen, expanded the literature cited and have presented new data to validate the phenotypes generated by their hit.

In general, this paper presents a refined method for using zebrafish as a model for segmentation defect.

No response necessary.